# Opening the Black Box of Imputation Software to Study the Impact of Reference Panel Composition on Performance

**DOI:** 10.3390/genes14020410

**Published:** 2023-02-04

**Authors:** Thibault Dekeyser, Emmanuelle Génin, Anthony F. Herzig

**Affiliations:** 1Inserm, Université de Brest, EFS, UMR 1078, GGB, F-29200 Brest, France; 2CHRU Brest, F-29200 Brest, France

**Keywords:** genotype imputation, population genetics, rare variants, reference panel, admixture

## Abstract

Genotype imputation is widely used to enrich genetic datasets. The operation relies on panels of known reference haplotypes, typically with whole-genome sequencing data. How to choose a reference panel has been widely studied and it is essential to have a panel that is well matched to the individuals who require missing genotype imputation. However, it is broadly accepted that such an imputation panel will have an enhanced performance with the inclusion of diversity (haplotypes from many different populations). We investigate this observation by examining, in fine detail, exactly which reference haplotypes are contributing at different regions of the genome. This is achieved using a novel method of inserting synthetic genetic variation into the reference panel in order to track the performance of leading imputation algorithms. We show that while diversity may globally improve imputation accuracy, there can be occasions where incorrect genotypes are imputed following the inclusion of more diverse haplotypes in the reference panel. We, however, demonstrate a technique for retaining and benefitting from the diversity in the reference panel whilst avoiding the occasional adverse effects on imputation accuracy. What is more, our results more clearly elucidate the role of diversity in a reference panel than has been shown in previous studies.

## 1. Introduction

Imputation of missing genotypes is a widely used technique for enriching genetic datasets where strong patterns of linkage disequilibrium between physically close genetic variants facilitate the highly accurate inference of missing data points [1]. Leading algorithms for genotype imputation are based on the Li-Stephens haplotype mosaic model [2] and are implemented with increasingly stream-lined Hidden Markov models (HMMs) [3,4,5,6]. Typically, when performing imputation, two datasets will be in play: a group of ‘target’ individuals for whom some genetic data are available and a panel of ‘reference’ individuals who have both genetic data in common with the target group as well as additional data for other genetic variants for which the genotypes of the target group are unknown. By comparing the target and reference individuals across the set of genetic variants that are present in both datasets, imputation algorithms attempt to infer the genotypes of the target individuals for the genetic variants that have only been measured in the reference panel. 

Globally, the literature is well stocked regarding studies of genotype imputation accuracy, with comparisons of different software [7,8], the impact of the choice of reference panel [9,10,11], or the type of genetic data involved [12]. Studies that have examined the choice of reference panel have broadly come to a consensus in that imputation will be more accurate when using a reference panel that is closely matched in terms of ancestry to the target group [13,14,15,16]. Yet, it has also been noted that having a reference panel that contains individuals from diverse populations is also beneficial [17,18,19]. The idea is that ‘unexpected-sharing’ [17] will improve imputation and that the HMMs used in imputation algorithms will only use the diversity in the reference panel of haplotypes when necessary. Therefore, the general advised strategy for imputing target individuals from a given population is to employ a reference panel that contains individuals of the same population along with individuals from neighbouring populations and even from more distant populations (in terms of the number of generations to trace back before common ancestors could be found for pairs of individuals) [20,21,22,23,24,25,26]. Simply put, the imputation panel cannot be too large, adding additional samples should only improve imputation, and the results presented for the largest and most cosmopolitan panels are certainly convincing [11,27]. 

In this study, we focus on the choice of reference panel but rather than simply measuring which reference panel provides the most accuracy, we explore, in greater detail, the role of reference panel composition in the context of large cosmopolitan panels. This is achieved by tracking, for each target individual and each genomic region, the ancestry groups of the reference panel individuals used in the imputation. This sheds light on previously published observations of certain imputation panels out-performing others, as well as indicating potential avenues for improving the performance of existing imputation algorithms. We find that more diversity may indeed improve the imputation of rare variants but that there may also be many imputed ‘false positives’ (incorrectly imputed genotypes containing rare alleles where, in fact, none are truly present), and we illustrate how this might be avoided. 

We used leading imputation software IMPUTE5 [3] and MINIMAC4 [5] and considered only the most frequent imputation scenario where target individuals have single-nucleotide polymorphism genotyping array (SNP array) data and the reference panel individuals have whole-genome sequencing (WGS) data.

## 2. Materials and Methods

First, some background on genotype imputation methods is required. The Li-Stephens model describes a process where, given a large enough sample of N haplotypes from a population, an $N+1^{th}$ haplotype can be closely approximated as a mosaic of small haplotype segments or ‘chunks’ from the pool of N haplotypes. Imputation algorithms apply this through Hidden Markov modelling; each haplotype in the target group (the target data will be phased statistically or directly using family data) will be imputed, in turn, with the idea being to infer a mosaic of reference panel haplotypes that will approximate each target haplotype. As the reference panel will contain data on more sites than are known in the target group, the mosaics that are inferred will allow for the imputation of missing genotypes. The HMMs involved will have, as hidden states, the index of the reference panel haplotypes who are donating a mosaic segment at a set of points across the genome. This will be the set of genetic positions that are present in both the target dataset and the reference panel. The observed states are the phased genotype data of the target haplotypes with emission probabilities allowing for differences between the donating reference haplotypes and the observed genotypes that could arise from recent mutations or genotyping errors. Crucially, software such as IMPUTE5 and MINIMAC4, will not give details about exactly which reference panel haplotypes are donating at different points in the genome; they are rather black box like in nature. 

For each genomic position in common between the target and the reference panel, imputation software will assign a posterior probability for each haplotype in the reference panel being the donating haplotype for the mosaic being built. When imputing target haplotype j, $h_{i}^{j}$ will be the index of the donating haplotype at genomic site i (where i goes from 1 to S) among the N reference haplotypes, so taking a value k between 1 and N. The HMM will provide, via the forward–backward algorithm [28,29], the posterior probabilities for the different possible values of k for $h_{i}^{j}$, based on the observed alleles of haplotype j, which we denote simply as $o^{j}$ to represent the sequence of observed alleles, $o_{1}^{j},\;o_{2}^{j}$, …, $o_{S}^{j}$. We can denote these posterior probabilities as $P\left(h_{i}^{j}=k\;|\;o^{j}\right)$. In practice, $o^{j}$ will be a sequence of zeros and ones where zeros correspond to reference alleles and ones to the alternative alleles. Similarly, the reference panel haplotype data can be written as $H^{k\;}$ (for the $k^{th}$ reference haplotype), which is similarly a sequence of zeros and ones but across a larger set of genomic sites than for the target data. 

Through linear interpolation between adjacent sites, these posterior probabilities are approximated for sites that are not present in the target data; we denote such a site as $i^{\prime }$. If a perfect mosaic is found, only one such posterior probability would be non-zero and only one reference panel haplotype would be donating; hence, the imputed value for site $i^{\prime }$ would be $H_{i^{\prime }}^{k^{*}}$, a zero or a one, depending on the known allele in the sole donating reference haplotype k* (k* denoting the sole value of k for which $P\left(h_{i^{\prime }}^{j}=k\;|\;o^{j}\right)=1$). However, there may be multiple possible mosaics and so multiple values of k for which $P\left(h_{i^{\prime }}^{j}=k\;|\;o^{j}\right)$ is greater than zero, and so rather than imputing missing genotypes from just one reference haplotype, a weighted sum, or ‘dosage’, $d_{i^{\prime }\;}^{j}$ is given: $$d_{i^{\prime }\;}^{j}=\overset{N}{\underset{k=1}{\sum }}P\left(h_{i^{\prime }}^{j}=k\;|\;o^{j}\right)H_{i^{\prime }}^{k}$$

IMPUTE5 and MINIMAC4 give the values of $d_{i^{\prime }\;}^{j}$ from which the values of $P\left(h_{i^{\prime }}^{j}=k\;|\;o^{j}\right)$ cannot be recovered by the user. We observed that if $d_{i^{\prime }\;}^{j}$, is non-zero at a position $i^{\prime }$ that is observed to be a singleton in the reference panel, one could infer that the corresponding posterior probability of the reference haplotype carrying the singleton must also be non-zero. This led to the idea of simply injecting a lot of artificial singletons into the reference panel to track the imputation. Given that the algorithms of current software have become increasingly complex in a bid to achieve superior efficiency for analysing huge bio-bank-scale data; this was, for us, a more feasible idea than attempting to unpick the existing code or to re-implement the HMMs ourselves. Tracking the role of each reference haplotype in such a way would not be impossible but would, however, incur a certain computation burden, we, therefore, set about tracking groups of reference haplotypes.

We took data from the 1000 Genomes Project [30] (1000G) as our sandbox. This dataset contains 2504 individuals grouped into 26 populations who are themselves grouped into five super-populations (Appendix A): AFR, AMR, EAS, EUR, and SAS. We used the version made available here https://bochet.gcc.biostat.washington.edu/beagle/1000_Genomes_phase3_v5a/b37.vcf/ (accessed on 28 May 2019) by the authors of Beagle [4], a version with no variants with a minor allele count below 5, which greatly reduces the size of the data and enabled us to carry out our analyses in a respectable timeframe. We only considered bi-allelic variants for this study and, hence, removed multi-allelic positions from the 1000G reference panel. We only analysed the 22 autosomal chromosomes.

We decided to separate three populations (ACB, ASW, and MXL) and impute the 221 individuals from these populations (our target group) with a reference panel formed by the remaining 23 populations. We wanted to choose a target group of admixed individuals so reference haplotypes from different populations would be required for imputation. These three populations were chosen because ACB and ASW are known to be recently admixed and we wished to include a third population with different characteristics to ACB and ASW; MXL was selected based on observing the unsupervised admixture plots in the 2015 1000G publication [30], where, arguably, all AMR populations appear to show admixture but MXL in particular had a profile where two components were both giving large contributions and so it was chosen for our experiment. A principal components analysis of the 1000G data is given in Appendix A with the positions of ACB, ASW, and MXL highlighted. For the three target populations, we separated the sites present on the UK-biobank genotyping array (https://biobank.ndph.ox.ac.uk/showcase/refer.cgi?id=149601, accessed on 1 February 2022). These 715,454 sites (Figure 1) would be supplied to the imputation software; the other sites would be retained in order to assess imputation accuracy. In order to track which reference haplotypes were being called on to impute the individuals of our target group across the genome, we injected completed synthetic variants into the imputation reference panel. These variants would serve as indicators for each of the five reference groups; a synthetic variant tagging the EUR population would be 0 (the ‘reference’ allele) for all non-EUR haplotypes and 1 (the ‘alternative’) for all EUR haplotypes. Hence, in this example, when interrogating the imputed dosage data for this synthetic variant, a dosage of 1 would indicate that only EUR haplotypes were used to form the mosaic to achieve the imputation; a dosage of 0 would indicate, conversely, that only non-EUR haplotypes were used; and a value between 0 and 1 would show that both EUR and non-EUR haplotypes were used. 

To add synthetic variants systematically, we selected locations to tag in the following manner: among the genomic sites in common between the target group and the reference panel, i.e. the genomic positions on the UK-biobank array, we kept those with a minor allele frequency (measured across the whole of the 1000G) above 0.2 and thinned with a strict pruning ($r^{2}<0.02$). We then retained from this list those sites which were not ‘shoulder-to-shoulder’ to another variant in 1000G, i.e., there was no other variant in 1000G at a distance of 1 base pair. The idea was that we wanted there to be a gap (no other bi-allelic variant present in 1000G) 1 base pair downstream of our tagged variants so that we could add our synthetic variants. This step removed 637 potential tagged variants and gave us a final list of 32,279 variants. We then added five synthetic variants 1 base pair downstream of each of these tagged variants, one to track each of the five super-populations in the reference panel. We verified that adding these synthetic variants in batches of five (and with the five variants in each batch all sharing the same physical position) had no effect on the imputation of IMPUTE5; the same output was given with and without the synthetic variants. We refer to each batch of five synthetic variants as an ‘imputation barcode’.

Having added our 32,279 imputation barcodes, imputation was completed using IMPUTE5. We could then calculate the cumulative contributions of each of the five reference groups to the imputation using the imputation barcodes. This was compared to similar estimations using the chromo-painting [31] functionality of pbwt [32], supervised ADMIXTURE [33] (which required us to filter by MAF (>5%) and to perform LD-pruning (--indep-pairwise 50 10 0.1) in plink [34]) and SOURCEFIND [35] (which used the pbwt chromo-painting output between all 2504 individuals of 1000G as input). We conducted a second imputation where each individual was imputed with a reference panel consisting of the super-populations that had a SOURCEFIND proportion above 0.01 for the individual in question. Data manipulation was performed using the R-package ‘gaston’ [36], shapeit2 [37], and bcftools [38].

Imputation accuracy was assessed by calculating the aggregate $R^{2}$ (AR2) [27] across difference sets of variants depending on the minor allele count (MAC) in either the 1000G as a whole or measured in each of the three target groups. Finally, we explored the impact of reducing the size of the reference panel based on an informed choice after examining the imputation barcodes (‘informed choice’ strategy). In order to track the imputation at the haplotype level (and not at the individual level), we imputed 442 (221 × 2) pseudo-individuals, which were simply each haplotype of the target group paired with itself. After imputation, we divided their dosages by 2 to give haplotype dosages and calculated genotype dosages by summing the two haplotype dosages of each individual. IMPUTE5 splits the autosomes into 400 imputation chunks and processes them separately. Chromosome 1 was split into 32 chunks, while chromosome 22 into only four chunks. For each chunk, we chose to impute each target haplotype with only a subset of the five super-populations. The choice was as follows: if none of the synthetic variants across all the imputation barcodes in the chunk that tag super-population ‘Z’ for haplotype j had a dosage above 0.9, then super-population ‘Z’ would be removed from the reference panel for haplotype j. Simply put, if on the first imputation run using all five super-populations, there was no evidence that super-population ‘Z’ was making a telling contribution somewhere in the chunk, we would remove it from the reference panel. Hence, each target haplotype would be assigned its own imputation panel for each of the 400 imputation chunks. For each individual, we summed the haplotype dosages to form genotype dosages after the separate imputation of each haplotype. The motivation for this procedure was to see if imputation accuracy would be affected by removing reference panel haplotypes that were only making a very small contribution, which we imagined might contribute more noise than it would improve imputation accuracy. Figure 1 synthesises our approach of adding synthetic variants and the resulting informed choice imputation strategy. 

## 3. Results

Having imputed the 221 target individuals using IMPUTE5, we summed and scaled the dosages of all imputation barcodes for each individual in order to provide an estimation of the contribution of each super-population to the imputation. In Figure 2, these proportions are compared to three other alternative estimations using software typically applied for the analysis of population genetics. 

The proportions derived from the imputation barcodes (Figure 2 top left) are, unsurprisingly, very similar to the chromo-painting-based estimates (Figure 2 top right), given that chromo-painting also invokes the Li-Stephens model. ADMIXTURE (Figure 2 bottom left) and SOURCEFIND (Figure 2 bottom right), however, give slightly different estimations; individuals from ACB and ASW are described as largely having AFR and EUR as source populations and MXL as having EUR and AMR. Indeed, the reader can compare these results to the unsupervised admixture plots in [30], which show very similar patterns for MXL, ACB, and ASW. What is different is that both our IMPUTE5-derived method and chromo-painting ascribe small but distinctly larger proportions to the other super-populations. Hence, when imputing these 221 target individuals, all the super-populations in the reference panel are being called upon; hence, the diversity in the reference panel seems to be important. To test this, we simply imputed each individual with the super-populations that SOURCEFIND indicates. Specifically, we imputed each individual with only the super-populations with a SOURCEFIND contribution above 0.01 (Figure 3). Imputation accuracy was summarized by aggregate $R^{2}$ statistics for different groups of variants depending on the minor allele count in the whole of 1000G (Figure 3b top) or in the three target sub-groups (Figure 3b bottom). Here, follow the details of the distribution of different reference panels used for the imputation in Figure 3: 88 individuals were imputed with a reference panel consisting of the EUR and AFR individuals, denoted as {EUR, AFR}, 45 were imputed with {EUR, AMR, AFR}, 38 with {EUR, EAS, AMR, AFR}, 14 with {EUR, EAS, AMR, SAS, AFR}, 10 with {EUR, SAS, AFR}, 8 with {EUR, EAS, AMR, SAS}, 6 with {EAS, AMR}, 4 with {EUR, EAS, AMR}, 3 with {EUR, AMR, SAS, AFR}, 3 with {EUR, EAS, AFR}, and 2 with {EUR, AMR}. In total, EUR was used for the imputation of 215 individuals, 201 for AFR, 120 for AMR, 73 for EAS, and just 35 for SAS.

Most individuals had an improved imputation accuracy when using the SOURCEFIND populations (Figure 3a), but the accuracy of imputation slightly suffered for rare variants (Figure 3b), particularly for individuals of MXL and particularly for variants with a low MAC in the 1000G as a whole. Only for very rare variants did the accuracy of imputation marginally decrease for the ACB and ASW individuals; elsewhere, it increased and, overall, the ASW and ACB groups showed noticeably different patterns to the MXL group. Indeed, using the populations indicated by SOURCEFIND worked well overall for ACB and ASW but much less well for MXL where individual aggregate $R^{2}$ did not increase in the same way as the ASW and ACB individuals and even a few individuals, in particular, had less accurate imputation (Figure 3a). The 14 individuals of MXL whose imputation accuracy fell noticeably are among the 20 individuals for whom the AFR super-population was not included in the reference panel. Rare variants were, unsurprisingly, less well imputed in general than common variants (Figure 3b, left panels). Another interesting observation was that depending on how the minor allele count was defined, it could easily either be concluded that the MXL group was imputed better or worse than the other two groups (ACB and ASW); note the difference in the x-axes between the top and bottom plots. In Appendix A, the individual AR2 and AR2* statistics from Figure 3a are given and, in fact, at the individual level, the MXL individuals have higher imputation accuracy. Note that, crucially, when regrouping variants by the minor allele count in each target group (bottom panels of Figure 3b), it is not possible to calculate the aggregate $R^{2}$ for variants that are truly monomorphic in the target group. However, the imputation of such variants may be incorrect; grouping variants by the MAC in the whole of 1000G allows such variants to be included.

To go further, we reasoned that whilst some improvement for rare variants was observed when imputing using a more diverse panel (notably for MXL in Figure 3 and even for ACB and ASW for very rare variants), this improvement might not be uniform across the genome. For example, individuals from ACB might benefit from reference panel haplotypes aside from those coming from EUR and AFR, but only sporadically. However, in many parts of the genome, imputing with a panel of only EUR and AFR would be at least equivalent and perhaps even more accurate given the results of Figure 3, where the majority of the individuals of ACB and ASW had a more accurate imputation with smaller less diverse imputation panels. We, here, attempt to demonstrate this in the following manner: for each imputation chunk (IMPUTE5 splits the autosomes into 400 chunks for imputation, as described in the Methods), we would impute each target haplotype with a specific reference panel, the choice of which was informed by the imputation barcodes in an initial imputation run using the whole reference panel (all of 1000G aside from the three target groups). This ‘informed choice’ strategy is outlined in the Methods and essentially represents a more fine-grained approach than simply using the SOURCEFIND proportions. The imputation accuracy from this strategy is compared to that of the initial imputation using all populations in Figure 4. Whilst the information from the imputation barcodes comes from IMPUTE5, we mirrored the initial imputation and the informed choice imputation with MINIMAC4 (Appendix A); the results were very much like those of IMPUTE5, showing that the patterns do not arise from some particularity in the IMPUTE5 software. Individual AR2 and AR2* statistics are given in Appendix A, where results were very similar between IMPUTE5 and MINIMAC4. We also observed that MINIMAC4 gave slightly higher accuracy for variants with a low MAC across the whole of the 1000G but slightly lower accuracy for common variants (Appendix A). Indeed, Appendix A again shows the impact for interpretation of the way that variants are grouped when calculating aggregate $R^{2}$ statistics.

The results from the informed choice imputation show that the same imputation accuracy can be achieved with a reduced reference panel, and the relative contributions of the different super-populations (Figure 4a, full comparison given in Appendix A) resembled something intermediate between our first computation and those attributed by SOURCEFIND (Figure 2). What is more, the imputation accuracy actually increases when using the reduced imputation panels of the informed choice strategy; unlike in Figure 3a, all individuals now have a better imputation accuracy (Figure 4a). Furthermore, even for rare variants, the imputation accuracy under the informed choice strategy is never worse than the baseline. To observe, in closer detail, this small gain, Table 1 provides full details of the imputation (using IMPUTE5) of singletons, doubletons, tripletons, and variants that are monomorphic in the whole target group (ACB, ASW, and MXL combined). The counts of true genotypes are compared to the counts of hard-called dosages (dosages that are rounded to the nearest value out of 0, 1, or 2). An equivalent table for MINIMAC4 is given in Appendix A where, again, MINIMAC4 showed slightly higher performance for the imputation of rare variants compared to IMPUTE5. 

In both Table 1 and Appendix A, it can be noted that for the monomorphic variants (MAC = 0), the informed choice method is improving the imputation by returning fewer ‘false positives’ (true genotype is AA but imputed dosage indicates Aa). For the variants with just 1, 2, or 3 observed alternate alleles in the target group, the improvement is less striking and comes more from ‘false negatives’ (true genotype is Aa and imputed dosage indicates AA). This illustrates the different appearances of the results in Figure 3 and Figure 4 depending on whether variants are grouped by the MAC in each target group or in the 1000G, as it is only in the latter case that the monomorphic variants are included in the calculation. 

## 4. Discussion

Here, we showed the value of tracking mosaics for understanding the performance of an imputation reference panel. Whilst our technique of adding imputation barcodes is rather unwieldy, it is relatively simple to perform and allows users interested in imputation methods to explore, in more detail, what it is going on under the hood of leading imputation algorithms. We entered into this work with the question as to why reference panels that are more diverse generally improve on smaller reference panels. We observed this by showing that rare variants were often imputed with greater accuracy when using all super-populations of the 1000G compared to when each individual was imputed with just the super-populations that seemed relevant determined by SOURCEFIND. Further, certain individuals were globally less well imputed (Figure 3a) with a less diverse reference panel, particularly a group of 14 from the MXL group for whom the AFR super-population was not strongly indicated by SOURCEFIND. This suggested that the more distant populations (those with small SOURCEFIND proportions), whilst only having a small genome-wide contribution, were sometimes improving imputation. We were able to show that we could essentially achieve equivalent imputation accuracy by leaving out reference panel individuals if there had not been evidence that they were making an important contribution in a given genomic region. This ‘informed-choice’ approach, in fact, even marginally improved imputation accuracy. We would not suggest that this method put forward here would necessarily be of use in practice as the improvements in accuracy are small and would, for example, only correspond to a very small gain in power for association testing. However, this work does help elucidate the performance of our reference panel and could provide avenues for improvements to existing imputation algorithms. It also challenges the concept that a reference panel can never be too large, at least when using current imputation software. Indeed, somewhat similar observations can be found in the literature of HLA imputation [39], a region that requires different imputation methods. 

The improvement attained by the informed choice strategy was most noticeable for rare variants in terms of the minor allele count in the 1000G. In Table 1, the marginal increase in accuracy could be observed notably for variants that were truly monomorphic in the target group; here, the informed choice method imputed over 70,000 fewer incorrect heterozygote genotypes compared to the baseline imputation. For variants with one, two, or three alternate alleles in the target group, the increase in accuracy comes instead from slightly fewer truly heterozygous genotypes being imputed as homozygous for the reference allele. We also saw that the impact of changing the reference panel was greatest for the MXL population. This population may present different challenges for imputation algorithms compared to ACB and ASW as the imputation of MXL seems to be relying on AMR haplotypes (of which there are fewer in the 1000G compared to AFR and EUR). The AMR group is also less well defined; many of the individuals have a European component (again, see unsupervised admixture plot in [30]). Indeed, the other populations of AMR have shown (in [30]) quite different admixture profiles and so the observations here of imputation performance for MXL would not necessarily transfer precisely to the other AMR populations. We observed that the added diversity of the reference panel may indeed help in the imputation of genotypes containing rare alleles (Figure 3) but, at the same time, the added diversity could lead to both false positives and false negatives (Figure 4 and Appendix A, Table 1 and Appendix A) that could otherwise be avoided. As matching between target and reference alleles is made at common variants, it is reasonable to imagine that when individuals from the target and reference group come from more distant populations, their most recent common ancestor might not be particularly recent, and so while they might share similar haplotypes for common (older) variants, there may be less similarity for rarer (more recent) mutations. Hence, we would propose that when the HMM encounters cases of ambiguity and several reference haplotypes will contribute to the final dosage, imputation might be slightly improved by giving priority to the haplotypes from the reference groups that have proved more widely relevant to the target haplotype across the genome. Such an approach would likely necessitate two passes of the HMM but given the high performance (in terms of speed and memory usage) that imputation algorithms have attained, this would not come at too high a cost. Indeed, a somewhat similar idea has already been put in place for combining imputation outputs when different reference panels have been used [40]. This meta-imputation technique also relies on multiple initial imputations with a second pass of an HMM to combine the inference of the different imputation realisations.

One aspect that we have not investigated is the choice of genotyping array. The UK biobank array is relatively dense and includes many variants that were selected to improve imputation accuracy, albeit based on imputation studies in European populations [41]. Using a far denser genotyping array could aid the imputation algorithms to form more precise mosaics and, hence, avoid some of the false positives observed in this study. Similarly, a less dense genotyping array would have likely performed worse and perhaps led to more false positives when using a diverse reference panel. Indeed, the results shown here could motivate the need for genotyping arrays that are specifically designed for imputation (perhaps of admixed individuals) when highly diverse reference panels are to be used.

Setting aside whether or not the ideas put forward here could practically be used to improve imputation accuracy, the results presented here certainly shed light onto the performance of imputation panels. The diversity in the sample was shown to contribute to the imputation with both positive and negative impacts. Simply removing diversity from the reference panel based on SOURCEFIND resulted in a loss of accuracy for the imputation of rare variants in the MXL group and for some individuals as a whole. However, a more diverse panel was also shown to lead to reduced imputation accuracy, in particular leading to many false positives, when compared to our informed choice strategy. This suggests that there could be situations where less could be more when choosing an imputation reference panel. 

## Figures and Tables

**Figure 1 genes-14-00410-f001:**
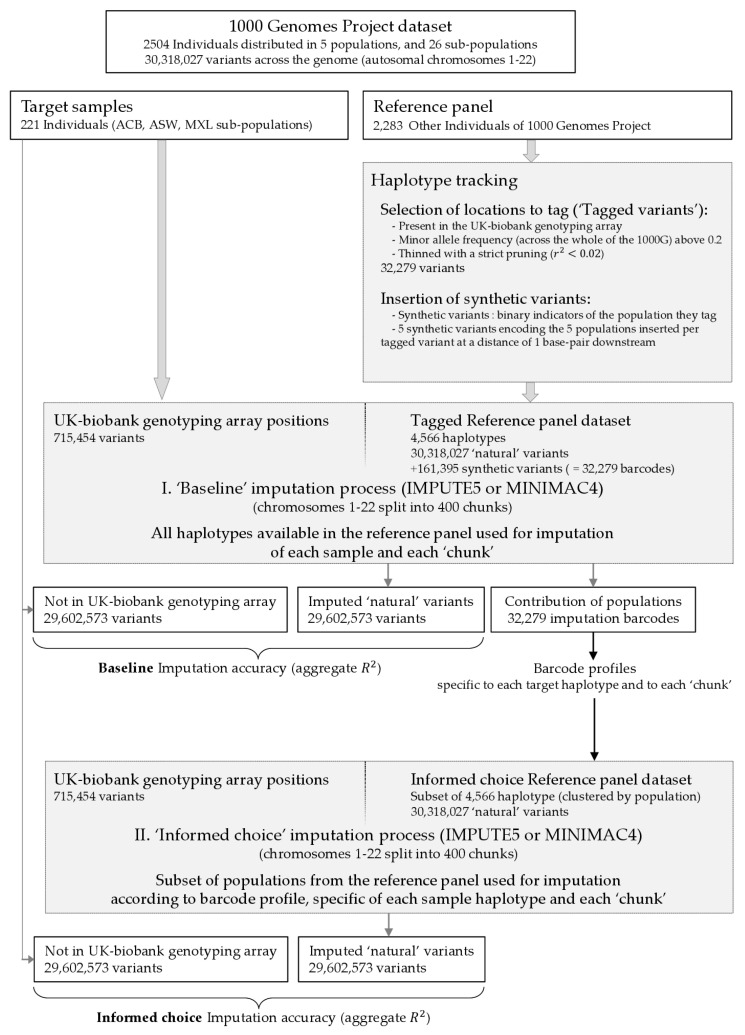
Schematic of the insertion of synthetic variants (imputation barcodes) and the informed choice imputation strategy.

**Figure 2 genes-14-00410-f002:**
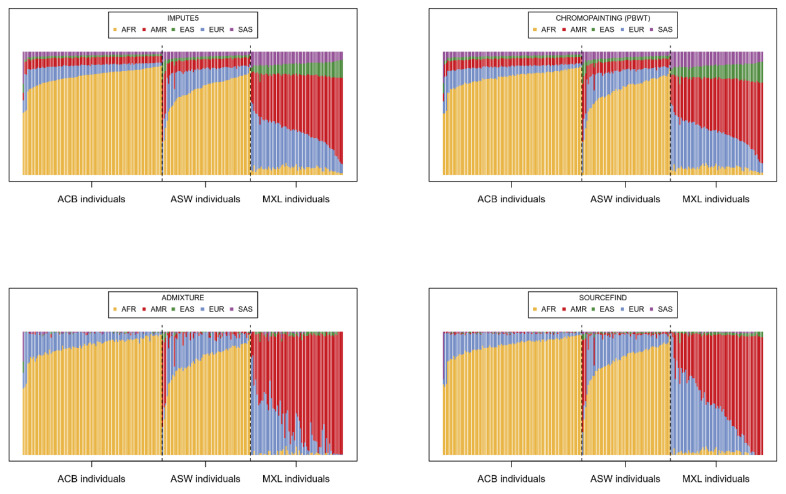
In these plots, each of the 221 target individuals has a vertical bar, which is coloured depending on the proportion of their genome assigned to the different super-populations from the reference panel. The individuals are in the same order across the plots, arranged according to the proportions in the top left plot. Top left: Proportions of the genome (autosomal chromosomes) imputed with haplotypes from the 5 super-populations AFR, EUR, EAS, SAS, and AMR, as ascertained by cumulating the dosages of the imputation barcodes. Top right: Proportions are estimated from the total ‘chunk length’ matrix derived from the chromo-painting algorithm of software pbwt. Bottom left: Ancestry proportions assigned by applying ADMIXTURE (supervised mode). Bottom right: Proportions assigned by SOURCEFIND.

**Figure 3 genes-14-00410-f003:**
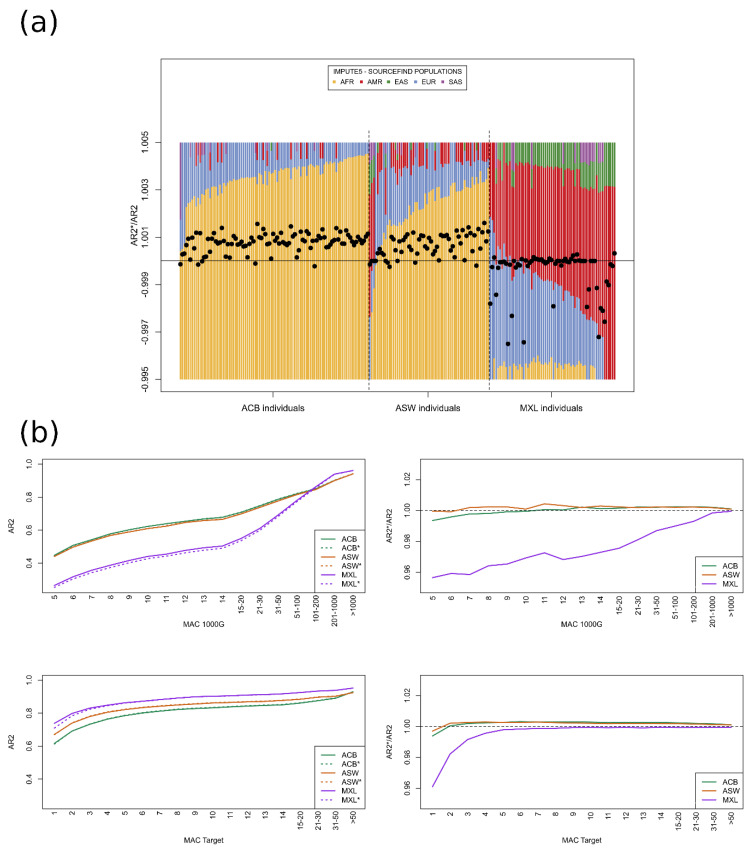
(**a**) The plot of proportions is calculated from the cumulated imputation barcodes after we selected a reference panel for each individual based on the SOURCEFIND proportions. Here, the individual level AR2*/AR2 are overlaid for each individual. AR2*/AR2 refers to the ratio of the aggregate $R^{2}$ using the reduced reference panels based on SOURCEFIND (AR2*) over the base line aggregate $R^{2}$ when all super-populations are used (AR2). Points above the horizontal line at AR2*/AR2 = 1 show that the individual’s imputation accuracy improved when only using the super-populations with a SOURCEFIND proportion above 0.01. (**b**) Here, the aggregate $R^{2}$ (AR2) statistics of imputation using all 23 non-target populations of 1000G and IMPUTE5 are compared to the aggregate $R^{2}$ when only the super-populations with a SOURCEFIND proportion above 0.01 were used. The results are split by population (ACB, ASW, and MXL) with lines marked with and without a ‘*’ corresponding to AR2* and AR2, respectively. On the two left panels, the aggregate $R^{2}$ statistics are split by minor allele count (MAC) bins where MAC is either calculated in 1000G (top) or separately in each of the three target sub-groups (bottom). The 1000G dataset used contained variants with an MAC of at least 5, hence the difference in the x-axes between the top and bottom panels. As the aggregate $R^{2}$ statistics are so close, we also show their ratio (AR2*/AR2) (right two plots).

**Figure 4 genes-14-00410-f004:**
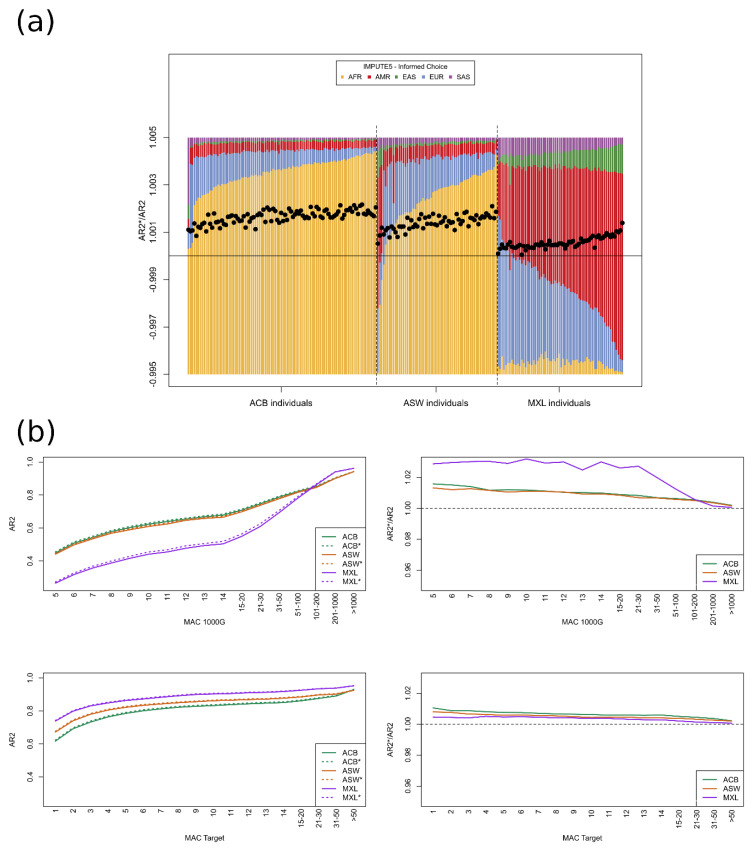
Similar to Figure 3, here, we compare the ‘informed choice’ strategy with the base line imputation. (**a**) The proportions plot corresponds to the cumulative proportions across imputation barcodes under the ‘informed choice’ strategy. The points overlaid correspond to individual AR2*/AR2 statistics as in Figure 3, but the AR2* now comes from the ‘informed choice’ strategy. (**b**) As Figure 3b but again the AR2* statistics come from the ‘informed choice’ strategy.

**Table 1 genes-14-00410-t001:** Details of the imputation of genotypes for variants which were truly monomorphic in the complete target group, or had a minor allele count (MAC) of one, two, or three in the complete target group (ACB, ASW, and MXL combined). The table compares true genotypes with hard-called imputed genotypes when either imputing with IMPUTE5 and the whole of the 1000G (minus the three target populations) as a reference panel, or with the same software and reference panel but when each target haplotype was imputed with a chosen subset of reference haplotypes informed by the imputation barcodes (‘informed choice’). For the ‘informed choice’ columns, each cell also includes the percentage increase or decrease compared to the corresponding cells when using all populations. AA refers to homozygous-for-the-reference genotypes, Aa for heterozygous genotypes, and aa for homozygous-for-the-alternative genotypes. Here, MAC is measured across the whole target group: ASW, ACB, and MXL together.

Hard-Called Dosage →		IMPUTE5			IMPUTE5, Informed Choice		
Truth ↓		AA	Aa	aa	AA	Aa	aa
MAC 0	AA	1437998689	920796	907	1438068970 + <0.01%	850635 −7.62%	787 −13.2%
	Aa	-	-	-	-	-	-
	aa	-	-	-	-	-	-

MAC 1	AA	626631509	738813	818	626633230 + <0.01%	737167 −0.22%	743 −9.17%
	Aa	929988	1919673	2026	915589 −1.55%	1934109 +0.75%	1989 −1.83%
	aa	-	-	-	-	-	-

MAC 2	AA	466448959	681496	618	466443614 − <0.01%	686783 +0.78%	676 +9.39%
	Aa	1208483	3032660	3071	1183935 −2.03%	3057009 +0.80%	3270 +6.48%
	aa	1862	3473	5527	1815 −2.52%	3461 -0.35%	5586 +1.07%

MAC 3	AA	343276195	601630	743	343270757 − <0.01%	607094 +0.91%	717 −3.50%
	Aa	1203342	3482444	4509	1176800 −2.21%	3508873 0.76%	4622 2.51%
	aa	2831	6730	11285	2755 −2.68%	6595 −2.01%	11496 +1.87%

## Data Availability

All data and software used here are publically available. The 1000 Genomes data were downloaded from here: https://bochet.gcc.biostat.washington.edu/beagle/1000_Genomes_phase3_v5a/b37.vcf/, accessed on 28 May 2019. Scripts for reproducing all results will be gladly shared on request and will be made publicly available here https://github.com/a-herzig/ImputationBarcodes, accessed on 28 May 2019.

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
