# Peer review of "Opening the Black Box of Imputation Software to Study the Impact of Reference Panel Composition on Performance"

_genes, 2023, doi:10.3390/genes14020410_

Round 1

Reviewer 1 Report

The manuscript entitled “Opening the black-box of imputation software to study the impact of reference panel composition on performance” by Dekeyser et al. describes an attempt to reveal the extent of contribution of ethnic diversity of the genome reference panels in the genotype imputation by the use of commonly available software. The authors neatly placed population barcodes into the each subpopulation of the International 1000 genome project reference panel and checked the contribution of subpopulations for the mosaic buildings of the haplotypes by the processes of imputation. The authors revealed that incorrect genotypes may be imputed if more diverse haplotypes were included in the reference panel. Taking into the consideration of the contributions of an ethnic subpopulation in the mosaic haplotypes in the imputation process, by the use of their population barcodes, they proposed an improved method, informed choice of reference panel, for imputation. The readers could understand the background of the imputation process through the paper and get some insight that the effect of the diversity of the reference panel in imputation process.

Minor points

!. It is better to show a scheme (flow chart) for explanation of the process they did in the manuscript. All the methodological process were described in the main text but it is somehow unfriendly to the novice readers.

2. In the methods section, the authors removed the “shoulder-to-shoulder” variants from the reference panel. It sounds like the most of, if not all, removed variants actually consist of the multi-nucleotide variants. So, the reviewer proposes to amend the description as “We then removed potential multi-nucleotide variants in 1000G”.

Author Response

Thank you kindly for your comments and suggestions that have helped us clarify the manuscript greatly, here follows our replies to your points raised:

1. It is better to show a scheme (flow chart) for explanation of the process they did in the manuscript. All the methodological process were described in the main text but it is somehow unfriendly to the novice readers.

Thank you that is very good suggestion, we have added a new Figure 1 with a flow chart explaining the process.

  1. In the methods section, the authors removed the “shoulder-to-shoulder” variants from the reference panel. It sounds like the most of, if not all, removed variants actually consist of the multi-nucleotide variants. So, the reviewer proposes to amend the description as “We then removed potential multi-nucleotide variants in 1000G”.

Thank you for pointing this out and we have made a few revisions to the methods to explain this part. Indeed we actually removed all the multi-allelic variants at the very beginning but we omitted to say that (this has been rectified). The shoulder-to-shoulder variants are a small number of sites where there are 2 SNPs in the 1000G 1 base-pair apart. As we wished to place synthetic variants 1 base-pair downstream of a tagged variant, we had to avoid such sites. This led to us excluding 637 potential tagged variants - and this is now detailed in the main text.

Reviewer 2 Report

Dekeyser and colleagues present a study exploring in what extent SNP imputation can be impacted by ancestry modification of reference panel in admixed populations. They used an original method to assess which 1KG superpopulation bears the largest contribution to local chunk of imputed haplotypes: informed choice strategy. The manuscript is well written and brings some interesting information about SNP imputation, however, it requires some clarification. Please, find specific comments below:

Major:

1.     Method. What the authors mean by “the more admixed individuals of 1000G” (line 140)? Is there a metric to determine this? I would say all AMR population are rather admixed ones, not only MXL. Maybe showing a PCA plot in supplemental would allow to better visualize the different admixed situation. The other AMR populations show very different pattern of admixture and could convey different conclusion about this superpopulation. Also, why removing all 3 populations of interest out of the imputation reference panel and not using a leave-one-out method? This should be discussed.

2.     Method. The coverage of the UK-biobank genotyping array should be mentioned. This coverage has probably a major impact on the imputation results compared to higher coverage chip array. The datasheet of the array mentions a SNPs selection of >245k markers especially focusing on European ancestry ones. This could also have an influence on the imputation results. While it is understandable that not all chip array can be tested, these crucial information should be mentioned and discussed.

3.     Results. The paragraph starting at line 250 referring to the figure 2 does not describe well the showed results. The MXL results are very different from the other 2, the authors seem to try to minimize this fact. Most individuals from ACB and ASW showed imputation improvement while most MXL individuals did not, as presented by the bottom right figure in 2b. I would advise to modify this paragraph.

4.     Results. The x axes of figures 2b and 3b are not at the same scale (starting at 1 or at 5) making it difficult to directly compare. Is a MAC of 5 in the full 1000G is comparable to a MAC of 1 in ACB, ASW or MXL?

5.     Results. Authors mentioned “The 14 individuals of MXL whose imputation accuracy fell noticeably among the 20 individuals for whom the AFR super-population was not included in the reference panel.”. Did the authors tried to add this population back to the reference panel or is it included in the informed choice strategy?

6.     Results. It is not clear to me how haplotypes are tracked and depicted in Figure 3. After using the haplotype information, the individual is reconstructed (or averaged) to make Figure 3?

7.     Results. Table 2. How the different genotypes were counted? Is it the sum of all individuals from the 3 population of interest?

8.     Results. Table 2 and S1. Authors should mention the higher performance of minimac4. Is it also the case for higher MAC? I guess this is well known?

9.     Results. Is there any specific chunk with better or worse imputation accuracy? Are the chunks including highly polymorphic regions (MHC, TCR, BCR, KIT) more difficult to impute? Is there an overrepresentation of a specific population in a given chunk?

10.  Discussion. What remains not clear to me in the discussion is how much improvement the informed choice strategy would provide if implemented? Could the authors add some quantification on this matter?

Minor:

1.     There are multiple typos throughout the manuscript, for example: lines 9, 69, 179. Please, check the manuscript carefully.

2.     Table 1 is not very informative and could be moved to supplementary material.

Author Response

Thank you kindly for your detailed and constructive comments on the manuscript which have helped us improve the work a lot; here follows our replies to each of your points:

Major:

  1. Method. What the authors mean by “the more admixed individuals of 1000G” (line 140)? Is there a metric to determine this? I would say all AMR population are rather admixed ones, not only MXL. Maybe showing a PCA plot in supplemental would allow to better visualize the different admixed situation. The other AMR populations show very different pattern of admixture and could convey different conclusion about this superpopulation. Also, why removing all 3 populations of interest out of the imputation reference panel and not using a leave-one-out method? This should be discussed.

We agree, ‘more admixed’ was quite a clumsy description and we have changed that. We chose ASW and ACB as these are known example of admixture and we agree that any of the AMR populations could have also been considered as admixed (which we have now mentioned). We chose MXL based on the admixture plots in the original 1000 Genomes publication (2015) as they seemed to have a profile of approximately 50%-50% between a component prevalent in EUR populations and a component prevalent in AMR populations.

Regarding the idea of a leave-one-out method - we feel that whilst this would have likely led to slightly better imputation statistics (e.g. because the individuals of ACB could be imputed with the individuals of ASW) we do not think it would have impacted the comparisons between different imputation strategies and the patterns we showed. It could have led to the admixture-style plots being rather confusing as well as the donor haplotypes would not have been the same for each group.

However, the main reason was that in this study we had to perform a very large number of genome-wide imputations and manipulations of the reference panel vcf files (which was time and memory consuming). By treating ACB+ASW+MXL as one target group allowed us to group many imputation runs together (noting that imputation is done haplotype-by-haplotype so there was no problem to impute the three groups at the same time with different reference panels).

We have added a PCA plot of 1000G indicating the positions of the three populations in to the supplementary materials.

  1. Method. The coverage of the UK-biobank genotyping array should be mentioned. This coverage has probably a major impact on the imputation results compared to higher coverage chip array. The datasheet of the array mentions a SNPs selection of >245k markers especially focusing on European ancestry ones. This could also have an influence on the imputation results. While it is understandable that not all chip array can be tested, these crucial information should be mentioned and discussed.

This is very salient and we have added text to the discussion regarding this point (see lines 440-449). We do feel that the choice of the UKbiobank array will not have greatly affected the study - it is a relatively dense array shown to perform well for imputation - indeed if we had wanted to show our ‘informed choice’ strategy working better we could have chosen an array with a lower performance. The fact that there are many variants to facilitate imputation in European populations will have potentially made the imputation of European haplotype segments rather stronger than other segments but this shouldn’t particularly affect our conclusions as we were observing notably the potential for ’false positives’ to occur due to the inclusion of more distant populations in the reference panel (e.g. maybe the populations from Asia given that we were imputing ACB, ASW, and MXL). 

  1. Results. The paragraph starting at line 250 referring to the figure 2 does not describe well the showed results. The MXL results are very different from the other 2, the authors seem to try to minimize this fact. Most individuals from ACB and ASW showed imputation improvement while most MXL individuals did not, as presented by the bottom right figure in 2b. I would advise to modify this paragraph.

On re-reading we agree with the reviewer and we have re-written this paragraph as it was neither clear nor a good description of the results, thank you for pointing this out. See paragraph starting line 286.

  1. Results. The x axes of figures 2b and 3b are not at the same scale (starting at 1 or at 5) making it difficult to directly compare. Is a MAC of 5 in the full 1000G is comparable to a MAC of 1 in ACB, ASW or MXL?

MAC of 5 in 1000G translates to a MAF of 5/5004 (0.001). MAC of 1 in either ACB, ASW, or MXL translates to a MAF of roughly 1/200 (0.005). We have added text to explain that the aim of these Figures is not to be able to directly compare the imputation accuracy statistics - it was more to show that different tendencies could be observed depending on how the variants are binned for the calculation of aggregate R2 statistics. We could start the x-axis at 1 for the top plots but this would lead to a large blank space and might be even more confusing. Particularly as MAC=5 in the 1000G is actually a lower MAF than MAC=1 in a target group. Hence we have just modified the text in the figure legend and results section to explain a bit more carefully the plot.

  1. Results. Authors mentioned “The 14 individuals of MXL whose imputation accuracy fell noticeably among the 20 individuals for whom the AFR super-population was not included in the reference panel.”. Did the authors tried to add this population back to the reference panel or is it included in the informed choice strategy?

No, we did not try to directly add back AFR for these 14 individuals but we agree that this would have almost certainly restored the imputation accuracy. And yes, for the informed choice strategy these 14 individuals were able to benefit from the AFR populations during the imputation.

  1. Results. It is not clear to me how haplotypes are tracked and depicted in Figure 3. After using the haplotype information, the individual is reconstructed (or averaged) to make Figure 3?

Yes the individuals are reconstructed as each haplotype is imputed separately using different panels but then genotype dosages are formed by summing across the two haplotypes. This is exactly what happens in normal imputation - the only difference here is that the informed choice strategy allows for different reference panels to be used for the two haplotypes of a single individual. Thank you for pointing this out as we do agree that this is not obvious and that we overlooked to precise just this in the manuscript - this has been added to the methods section (lines 215-216).

  1. Results. Table 2. How the different genotypes were counted? Is it the sum of all individuals from the 3 population of interest?

Yes indeed, and we have made that clearer in the text.

  1. Results. Table 2 and S1. Authors should mention the higher performance of minimac4. Is it also the case for higher MAC? I guess this is well known?

We included minimac4 simply to demonstrate that the difference between the baseline imputation and the informed choice imputation was not just due to some particularity of the IMPUTE5 algorithm. In general, IMPUTE5 and MINIMAC4 have a very similar model and have often been shown to give very similar imputation accuracy; though yes in our experience MINIMAC4 often holds a very small advantage. We have included another supplementary figure (Figure S5) comparing the imputation accuracy of the two - indeed MINIMAC4 does slightly better for very rare-variants but in fact IMPUTE5 has slightly higher accuracy for more common variants. The plot again shows the importance of considering variants that are monomorphic in the sample compared to when only considering polymorphic variants.

  1. Results. Is there any specific chunk with better or worse imputation accuracy? Are the chunks including highly polymorphic regions (MHC, TCR, BCR, KIT) more difficult to impute? Is there an overrepresentation of a specific population in a given chunk?

It is hard to quantify one genomic region being imputed with better or worse accuracy as they will contain different distributions of rare and common variants and as you point out, some regions are highly polymorphic. Indeed regions such as MHC are best imputed with specific software and we feel it is beyond the scope of this work to investigate this question; whilst it would make for an intriguing future study. Indeed, here we are trying to show the general interest of knowing which populations are contributing to imputation at different regions of the genome and by proposing this approach it opens up the possibility for other researchers to investigate if particular regions of the genome are being imputed differently to others using the technique of adding synthetic variants. We did however take the time to check that the global patterns that we saw were not driven by a single chromosome (such as chromosome 6) and it was not the case.

  1. Discussion. What remains not clear to me in the discussion is how much improvement the informed choice strategy would provide if implemented? Could the authors add some quantification on this matter?

We have expanded on what we wrote in the discussion, the improvement seems to be very marginal in the imputation scenario presented here and the complexity and time required for the method would probably make it an impractical choice for a real data application. However, we feel that this work has gone a long way to explaining the performance of a diverse imputation panel and could lead for possible small improvements to existing imputation algorithms. The main advantage seemed to be that by sometimes reducing the reference panel size, we avoided imputing ‘false positives’, particularly for rare-variants. This would likely give a small increase in power for association testing particularly for rare-variant burden tests. In general as a rule of thumb, the R2 statistics for imputation accuracy can give an approximation of the loss of power for association tests using imputed data compared to if true genotypes were known. Hence, the small increases in imputation accuracy (up to 2% for rare-variants and even less for common variants) give the answer and show that really not a huge improvement would be gained, hence why we have not stressed that the informed choice strategy should be used in practice but that it was useful for illustrating how imputation algorithms are working and how the diversity in the reference panel is being drawn upon.

Minor:

  1. There are multiple typos throughout the manuscript, for example: lines 9, 69, 179. Please, check the manuscript carefully.

We have re-checked the entire manuscript and corrected several typos

  1. Table 1 is not very informative and could be moved to supplementary material.

We have moved it to the supplementary materials.